# LSTR: LONG-SHORT RANGE AGGREGATION FOR TRAJECTORY PREDICTION AT INTERSECTION SCENARIOS

## ABSTRACT

Trajectory prediction is crucial for practical applications, encompassing navigation for autonomous vehicles and the implementation of safety systems based on the Internet of Vehicles (IoV). Most existing methods significantly rely on comprehensive map information, employing robust rule constraints to incrementally predict trajectories within the driver's local decision-making context. However, in environments characterized by weak rule enforcement, such as urban intersections, these approaches neglect the disparity between the driver's global intentions and current behaviors.Recognizing the characteristics of intersection traffic flow—macroscopically organized yet microscopically disordered, exhibiting highly heterogeneous conditions—this paper presents a novel model termed Long-short Range Aggregation for Trajectory Prediction in Intersections (LSTR). This model anchors the vehicle's local decision-making process to long-range intentions. Specifically, LSTR predicts the vehicle's destination via a global intention inference module and models its long-range driving intentions through clustering to extract macroscopic traffic flow patterns. This long-range intention subsequently informs the short-range local interaction behaviors captured by the local behavior decision module. Ultimately, the fused features from these two modules are analyzed using a multi-modal decoder to interpret the various motion patterns, resulting in the trajectory prediction outcomes.We rigorously validate the proposed framework across multiple intersection scenarios utilizing real-world datasets, including inD, roundD, and a subset of WOMD. Experimental results indicate that our model surpasses several benchmarks.

## 1 INTRODUCTION

Trajectory prediction is crucial for autonomous driving and intelligent transportation, providing foresight for vehicle planning, safety, and traffic management. Existing methods have achieved success in highway and urban road applications Cong et al. (2023); Shi et al. (2024); Sun et al. (2022), often relying on lane-level high-definition maps to spatially constrain trajectories and make locally optimal decisions. These methods can be seen as "greedy algorithms under strong constraints." However, in weakly constrained environments like urban intersections, they underperform due to neglecting multi-level flow dynamics, leading to a disconnect between global intentions and local decisions.

The multi-tiered flow dynamics at intersections can be modeled as a superposition of macroscopic and microscopic traffic flows Zhao et al. (2023).Microscopic traffic flow involves complex interactions and behaviors of intersection users, characterized by disorder and heterogeneity. Each user navigates the intersection with different strategies based on real-time conditions, the behavior of others, and individual decision-making, such as alternating vehicle passage or pedestrians crossing unexpectedly. This dynamism, driven by individual choices and implicit interactions, makes microscopic traffic flow inherently difficult to predict. Macroscopic traffic flow at intersections is defined by a weakly regulated pattern, with the primary constraint being the "route choice"—the selection of entry and exit routes. Despite the complexity of microscopic behavior, macroscopic flow reflects collective tendencies, providing essential global prior information for trajectory prediction.

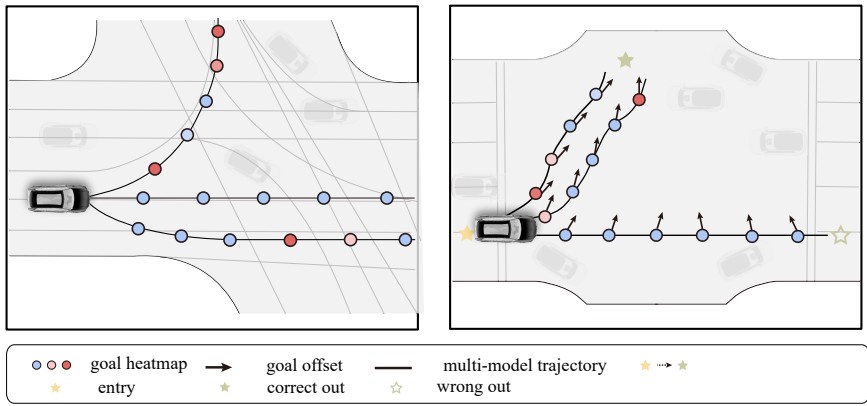

Figure 1: The left figure illustrates a greedy algorithm with strong rules, and it rely on lane-level high-definition maps to spatially constrain trajectories and make locally optimal decisions. While the right figure shows our LSTR method, we aim ta integrating local interaction modeling and global intention prediction within a framework

Accordingly, intersection trajectory prediction can be divided into two core challenges: local interaction scene perception and long-range global intention prediction. Local interaction scene perception primarily focuses on the dynamic coupling relationships within microscopic traffic flow. By capturing real-time interactions among traffic participants, it seeks to comprehend complex and heterogeneous traffic behaviors, thereby accurately modeling the mutual influence between individual vehicles. Global intention prediction, in contrast, centers on analyzing the global trends within macroscopic traffic flow. It relies on the relatively weak "route choice pattern" rule to discern and infer the overall driving intention of vehicles at intersections.

In this work, we present a novel trajectory prediction method for urban intersections, integrating local interaction modeling and global intention prediction within a Transformer-based framework. Our approach introduces two key innovations:(1) We propose a Local Behavior Decision Module (LBDM) to model local interactions. Temporal dependencies are captured through the Temporal Neighborhood Aggregation module, and short-range motion patterns are refined using the Coherent Occupancy Prediction (COP) head. This enables parallel prediction of future trajectories, effectively capturing local dynamics.(2) We propose a Global Intention Inference Module (GIIM), which predicts the destination, captures the "route choice pattern," and anchors global intentions with local decisions. These features are filtered and passed to the decoder, combining mode-specific and temporal encoding for path search, enabling accurate and interpretable predictions in complex intersections.(3) Using these techniques, the LSTR framework outperforms map-prior algorithms. In experiments on the inD, rounD, and WOMD datasets, where LSTR improves b-minFDE$_6$ on inD by 4.5 and minADE$_6$ on rounD by 4.2 over the second-best method.

## 2 RELATED WORKS

**Trajectory prediction based on HD maps.** Trajectory prediction is a well-researched area in autonomous driving Huang et al. (2023)Teeti et al. (2022), involving the estimation of future trajectories based on historical agent states and environmental information. HD maps play a critical role by providing detailed road structure, lane configurations, and traffic signs, significantly improving prediction accuracy Wang et al. (2023)Sharma et al. (2024). Methods leveraging HD maps are categorized into traditional kinematic models and deep learning approaches.Kinematic models use physics and kinematics to predict vehicle trajectories, considering position, velocity, acceleration, and road constraints Lefkopoulos et al. (2020)Okamoto et al. (2017)Wang et al. (2019)Zhang et al. (2017). While these models are interpretable and perform well in short-term predictions (up to one second), they struggle with complex maneuvers in dynamic environments like urban intersections due to simplifying assumptions.Recently, deep learning models have advanced trajectory prediction using

HD maps. Early methods Casas et al. (2020)Marchetti et al. (2020)Park et al. (2020)Zhang et al. (2020) encoded maps and agents into images using CNNs, followed by fully connected layers for prediction. Approaches like VectorNet Gao et al. (2020) and LaneGCN Liang et al. (2020) represent road structures and agent motion in graph networks to capture agent-map interactions. Target-driven models such as TNT Zhao et al. (2021) and DenseTNT Gu et al. (2021a) generate map-based candidate targets to regress trajectories. The MTR series Liu et al. (2021)Shi et al. (2024) uses learnable motion queries to search for strategies within lane-level constraints.While deep learning methods reliant on HD maps perform well, they falter in environments like urban intersections where map data is sparse, and vehicle strategies are influenced by global intentions and unstructured information (e.g., traffic flow, interactive behaviors, implicit rules). Greedy algorithms fail to model global strategies effectively, distorting predictions. Our approach explicitly models agents' global intentions through endpoint clustering, using this to constrain local interactions and ensure consistency between local decisions and global planning.

**Transformer-based Vehicle Motion Forecasting.** Transformers are a neural network architecture leveraging the attention mechanism, originally developed for machine translation in NLP, and have outperformed recurrent neural networks Brașoveanu & Andonie (2020). Recently, they have proven highly effective for trajectory analysis Quintanar et al. (2021). Huang et al. Huang et al. (2022) proposed a Transformer-based multimodal trajectory prediction model using multi-head attention to capture relationships between agents. SceneTransformer Ngiam et al. (2021) integrates features from agent interactions and road maps spatially and temporally, while LaneTransformer Wang et al. (2023) extends this with Attention-Based Block Aggregation for higher-order interactions.In summary, Transformers show great potential for trajectory prediction by capturing complex dependencies and interactions. They offer scalability, transfer learning, and the ability to manage multiple agents. Our LSTR approach builds on this by introducing a Transformer architecture that incorporates auxiliary tasks in both the local behavior decision module and global intent inference module. This design tightly constrains global intentions and local behaviors, enabling accurate and interpretable trajectory predictions in complex intersections.

# 3 APPROACH

**Problem Definition.** Intersection trajectory prediction aims to forecast future vehicle trajectories using historical data and map information. Mathematically, this is expressed as: $\widehat{Y}^i_{t+1:t+T} = f\left(X^i_{1:t}, M\right)$, where historical trajectories $X^i_{1:t} = \{(x^i_1, y^i_1), \ldots, (x^i_t, y^i_t)\}$ and map information $M$ are given. The goal is to predict $\hat{Y}^i_{t+1:t+T} = \{(x^i_{t+1}, y^i_{t+1}), \ldots, (x^i_{t+T}, y^i_{t+T})\}$ over the next $T$ steps. The model $f$ captures the relationship between local vehicle behavior and global driving intentions.

**Overview.** We propose a Long-short Range Aggregation Trajectory Prediction Model (LSTR) for urban intersections, utilizing a Transformer Encoder-Decoder framework to constrain multimodal future vehicle movements based on global intention. As shown in Figure 2(a), the LSTR comprises three modules. The Local Behavior Decision Module (LBDM) forecasts short-range dynamics using historical features, while the Global Intention Inference Module (GIIM) captures long-range motion patterns. These modules identify each vehicle's "path selection patterns." The multimodal decoder then dynamically aligns with the top-k optimal patterns and refines local decisions to predict multimodal trajectories.

**Backbone.** We vectorize input trajectories and intersection road maps as polylines (Gao, 2020) and encode the scene using an agent-centric approachGu et al. (2021b); Varadarajan et al. (2022); Zhang et al. (2020); Zhou et al. (2022). Our model uses the Encoder module from MTRShi et al. (2022) to process historical trajectory and map features. As shown in Figure 2(b), for each vehicle's historical trajectory $X^i_{1:t} = \{(x^i_1, y^i_1), \ldots, (x^i_t, y^i_t)\}$ and map information $M$, features are first extracted using a Multi-Layer Perceptron (MLP), followed by max-pooling to integrate local and global trajectory features. A spatial attention module then aggregates neighborhood features to capture vehicle-map interactions. After multiple iterations of the Encoder Block, the final feature representation $G = [F_A, F_M]$ is obtained, where $F_A = (N_a, D)$ and $F_M = (N_m, D)$ denote vehicle and map features, with $N_a$ and $N_m$ representing the number of vehicles and map polylines, respectively.

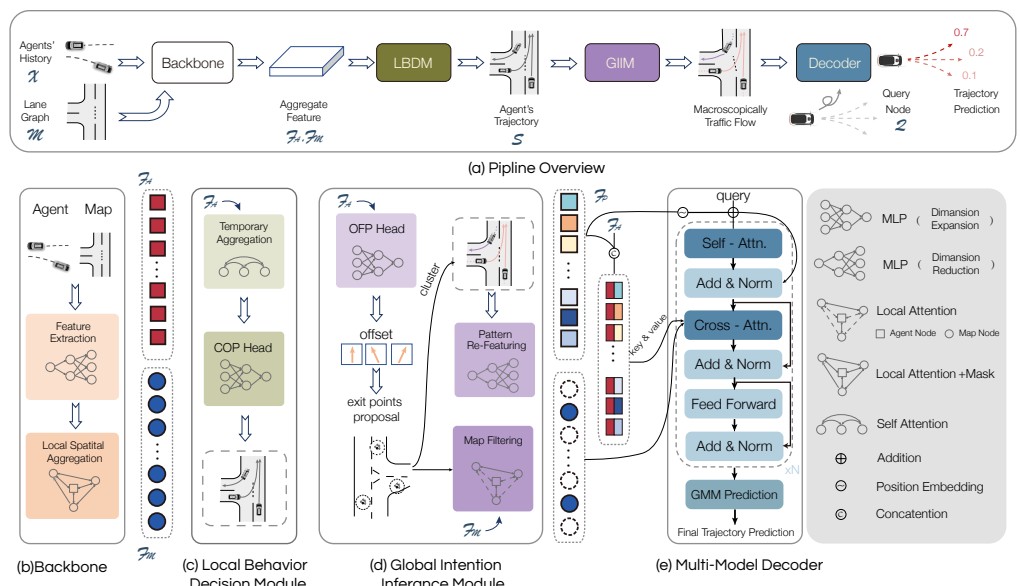

Figure 2: Overview of LSTR. LSTR takes as input the historical trajectory features of vehicles and surrounding road maps at intersections. The Backbone extracts spatially aggregated local features, while LBDM outputs local trajectories. GIIM then predicts destinations and generates global intentions. The Decoder uses an attention mechanism to filter the most likely global intention for each vehicle, producing multimodal trajectory outputs.

## 3.1 LOCAL BEHAVIOR DECISION MODULE (LBDM)

**Local Temporal Aggregation.** Capturing local interaction decisions between vehicles is crucial for trajectory prediction in microscopic traffic flow at intersections. Inspired by HiVTZhou et al. (2022), we use a multi-head attention mechanism based on the Transformer to model historical interactions and temporal dependencies, as shown in Figure 2(c). For each agent $i$, the aggregated trajectory feature at each historical time step $t' \in \{1, 2, \ldots, t\}$ is computed as $\mathbf{h}_{t'}^i = $ Concat $\left(\mathbf{h}_{t'}^{i,1}, \mathbf{h}_{t'}^{i,2}, \ldots, \mathbf{h}_{t'}^{i,H}\right) \mathbf{W}^O$ and $\mathbf{h}_{t'}^{i,h} = \sum_{j \in \mathcal{N}_i^{\tau}} \alpha_{t,h}^{ij} \mathbf{p}_{t'}^j$, where $\mathbf{h}_{t'}^i$ is the aggregated feature, $H$ is the number of attention heads, $\mathbf{W}^O$ is a learnable projection matrix, and $\mathbf{h}_{t'}^{i,h}$ is the output of the $h$-th attention head. The attention weights $\alpha_{t,h}^{ij}$, computed across neighboring trajectory points $\mathbf{p}_{t'}^j$ within the local temporal window $\mathcal{N}_i^{\tau}$, are obtained using scaled dot-product attention as $\alpha_{t,h}^{ij} = \frac{\exp\left(\mathbf{q}_i^h \cdot \mathbf{k}_j^h\right)}{\sum_{k \in \mathcal{N}_i^{\tau}} \exp\left(\mathbf{q}_i^h \cdot \mathbf{k}_k^h\right)}$, where $\mathbf{q}_i^h$ and $\mathbf{k}_j^h$ represent the query and key vectors of agent $i$ and its neighbor $j$, respectively.

**Coherent Occupy Prediction (COP).** The features encoded by the Backbone and temporal attention module $\mathbf{H}_{1:t} = \{\mathbf{h}_{t'}^i \mid i \in \{1, 2, \ldots, N\}, t' \in \{1, 2, \ldots, t\}\}$ capture the contextual historical information of all $N$ agents and their interactions with road elements. The COP Head estimates the 2D Gaussian distribution of occupancy and velocity for each vehicle over the next $T$ frames based on $\mathbf{H}_{1:t}$. Specifically, COP predicts $(\hat{\mu}_x, \hat{\mu}_y, \hat{\sigma}_x, \hat{\sigma}_y, \hat{\rho}, \hat{v}_x, \hat{v}_y)$ at each time step $t' \in \{t+1, \ldots, t+T\}$, where $(\hat{\mu}_x, \hat{\mu}_y)$ indicates the predicted position. This is compared with the actual trajectory $(x_{t'}^i, y_{t'}^i)$ using the L1 loss:

$$\mathcal{L}_{cop} = \frac{1}{N} \sum_{i=1}^{N} \sum_{t'=t+1}^{t+T} \left( \| \mathbf{X}_{t'}^i - \hat{\mu}_{t'}^i \|_2 + \lambda_v \| \mathbf{V}_{t'}^i - \hat{\mathbf{V}}_{t'}^i \|_1 \right) \tag{1}$$

where $N$ is the number of vehicles, and $T$ is the number of future time steps. $\mathbf{X}_{t'}^i = (x_{t'}^i, y_{t'}^i)$ and $\mathbf{V}_{t'}^i = (v_x, v_y)$ are the true position and velocity, while $\hat{\mu}_{t'}^i$ and $\hat{\mathbf{V}}_{t'}^i$ are predictions. COP

predicts both occupancy and motion, enabling the model to capture short-term dynamics and lay the foundation for subsequent tasks.

## 3.2 GLOBAL INTENTION INFERENCE MODULE (GIIM)

**Offset Flow Prediction (OFP).** The Offset Flow Prediction Head predicts directional offsets towards potential exit points at intersections, similar to Implicit Occupancy Flow FieldsAgro et al. (2023), focusing only on the queried continuous points' offsets.

As shown in Figure 2(d), the input to the Offset Flow Prediction Head is the trajectory feature $F_A$ from the Backbone network. For each agent $i$, instead of regressing absolute exit point coordinates, the Head predicts 2D offset vectors $\left\{\hat{o}^i_{t+1:t+T}\right\}$, representing displacements from predicted positions to exit points at future steps $t' \in \{t+1, \ldots, t+T\}$, defined as:

$$\left(\hat{o}^i_{x,t'}, \hat{o}^i_{y,t'}\right) = \left(e^i_x, e^i_y\right) - \left(\hat{p}^i_{x,t'}, \hat{p}^i_{y,t'}\right) \tag{2}$$

where $\left(\hat{p}^i_{x,t'}, \hat{p}^i_{y,t'}\right)$ are the predicted coordinates, and $\left(e^i_x, e^i_y\right)$ are the exit point ground truth coordinates, relabeled for each intersection scene. $\left(\hat{o}^i_{x,t'}, \hat{o}^i_{y,t'}\right)$ is the difference between them.

The loss function is the L1 loss between the predicted $\left\{\hat{o}^i_{t+1:t+T}\right\}$ and ground truth offset vectors $\left\{o^i_{t+1:t+T}\right\}$, calculated as:

$$\mathcal{L}_{\text{ofp}} = \frac{1}{N} \sum_{i=1}^{N} \sum_{t'=t+1}^{t+T} \left| o^i_{t'} - \hat{o}^i_{t'} \right| \tag{3}$$

Supervision is applied only on predicted positions, ignoring other locations.

**Macroscopic Traffic Flow Generation.** The predicted exit position for each vehicle is computed as $\left(\hat{e}^i_x, \hat{e}^i_y\right) = \left(\hat{\mu}_x, \hat{\mu}_y\right) + \left(\hat{o}^i_{x,t'}, \hat{o}^i_{y,t'}\right)$, where $\left(\hat{e}^i_x, \hat{e}^i_y\right)$ represents the predicted exit coordinates of agent $i$, while $(\hat{\mu}_x, \hat{\mu}_y)$ and $\left(\hat{o}^i_{x,t'}, \hat{o}^i_{y,t'}\right)$ are as defined in COP and OFP sections.

To determine candidate exit paths, we select predicted exit points with a Euclidean distance of less than 1 meter as candidates. If multiple points meet this condition, the geometric center is chosen to ensure uniqueness, and the process is illustrated in Figure 2(d). It gives $k$ candidate exits for all $N$ vehicles. The model is designed to learn macroscopic traffic flow patterns, so the matching between these $k$ candidate exits and $k$ ground-truth exits is treated as an assignment problemCarion et al. (2020), solved using the Hungarian algorithm. The distance matrix, based on Euclidean distance between predicted cluster centers $\hat{e}_i$ and actual exits $e_i$, defines the loss function:

$$\mathcal{L}_{\text{aux}} = \frac{1}{k} \sum_{i=1}^{k} \| \hat{e}_i - e_{\pi(i)} \|_2 \tag{4}$$

where $\| \cdot \|_2$ represents Euclidean distance, $\hat{e}_i$ is the predicted cluster center, and $e_{\pi(i)}$ is the corresponding ground-truth exit. Based on the learned exit and known entrance positions, clustering is performed using the "Intersection-Direction-Lane-Intention" keywordZhang et al. (2021) to obtain pseudo-labels for traffic flow. These pseudo-labels are converted to hidden features $F_P$ via an MLP and concatenated with trajectory features $F_A$, resulting in globally intention-anchored local features. Ablation experiments show that global intention prediction through cluster supervision improves convergence and model performance, due to the regularization effect of cluster supervision, which helps capture macroscopic traffic flow.

**Long-range Spatial Dependency Extraction.** Long-term temporal dependencies have been widely studiedChai et al. (2020); Lin et al. (2024); Ngiam et al. (2021), but effectively capturing long-range spatial dependencies remains underexplored. The entrances and exits of urban intersections, key constraints for trajectory prediction, contain the semantic information of the surrounding road environment. For each vehicle $i$, we filter the map features $F'_M$ around its predicted exit $\hat{e}_i$ and known entrance $s_i$, which are then input into the Decoder for trajectory prediction. Ablation experiments show that capturing long-range spatial dependencies enhances prediction performance.

### 3.3 MULTI-MODAL DECODER

The objective of this module is to predict $K$ multimodal trajectories for each target vehicle, completing the trajectory prediction task. The model has learned global intentions from previous modules, and vehicles with similar intentions have similar trajectories but differ in temporal shifts. To account for this, we introduce time-sensitive, learnable position embeddings to search for local trajectories across different modes. These embeddings are added to the global intention pseudo-labels $F_P$ and the Key and Value features of the target vehicles.

The multi-modal decoder first selects the $K$ mode labels $F_P'$ most similar to $F_P$ and adds them to the Query after position encoding to predict $K$ future modes. Key and Value consist of two parts: the vehicle's pseudo-label $F_P$, re-featurized and concatenated with the historical trajectory feature $F_A$, and the map features selected based on the predicted exit $\hat{e}_i$ and entry $s_i$. After position encoding, these are cross-attended with the Query, and the fused map and trajectory features are passed through an MLP into the prediction head. The prediction head outputs $K$ trajectories, each modeled by a 2D Gaussian distribution.

The predicted trajectory mean, standard deviation, and correlation coefficient capture trajectory uncertainty. The model is trained using negative log-likelihood (NLL) loss, optimizing the probability of the predicted distribution matching the real trajectory:

$$\mathcal{L}_{\text{NLL}} = -\frac{1}{N} \sum_{i=1}^{N} \sum_{t=1}^{T} \log p\left( \left( x_t^i, y_t^i \right) \mid \hat{\mu}_x^i, \hat{\mu}_y^i, \hat{\sigma}_x^i, \hat{\sigma}_y^i, \hat{\rho}^i \right) \tag{5}$$

where $p\left( \left( x_t^i, y_t^i \right) \mid \hat{\mu}_x^i, \hat{\mu}_y^i, \hat{\sigma}_x^i, \hat{\sigma}_y^i, \hat{\rho}^i \right)$ is the Gaussian probability density function, describing the difference between the predicted and true positions. Minimizing this loss allows the model to refine local decisions within global motion patterns for more accurate trajectory prediction.

## 4 EXPERIMENTS

### 4.1 EXPERIMENTAL SETTINGS

**Dataset.** The efficacy of our method has been validated on select intersection scenarios from the inD, rounD, and WOMD datasets. The inD dataset contains trajectory data for over 13,000 road users at four unsignalized urban intersections in Aachen, Germany, including 8,233 cars, trucks, and buses. The rounD dataset features motion trajectories for more than 13,700 road users from three roundabouts, where vehicles are less affected by VRUs due to the roundabouts' distance from city centers. Both datasets were captured via drones, with trajectories sampled at 25Hz. The Waymo Open Motion Dataset is the most diverse interactive motion dataset, comprising over 100,000 sequences and more than 200,000,000 frames of 3D maps and trajectory annotations, sampled at 10Hz. Detailed data preprocessing procedures are provided in the supplementary materials.

**Metrics.** We use the official Argoverse competition metrics:

- **minFDE$_k$:** the Euclidean distance between the endpoint of the best of $k$ predicted trajectories and the ground truth, averaged across all scenarios.
- **minADE$_k$:** the average Euclidean distance between the points of the best of $k$ predicted trajectories and the ground truth, averaged across all scenarios.
- **miss rate (MR$_k$):** the percentage of scenarios where the minimum final displacement error exceeds 2 meters.
- **b-minFDE$_k$:** the minFDE$_k$ with an added penalty based on the confidence $p$ of the best predicted trajectory, defined as $(1 - p)^2$.

The top-$k$ trajectories are those with endpoints closest to the ground truth. These metrics are computed for $k = 6$ and $k = 1$, except **b-minFDE$_k$**, which is only for $k = 6$. Note that our model predicts only vehicle trajectories and does not account for VRUs.

**Training Details.** Our model was implemented using the PyTorch and PyTorch-lightning deep learning frameworks on an experimental platform equipped with four A100 GPUs. The Adam

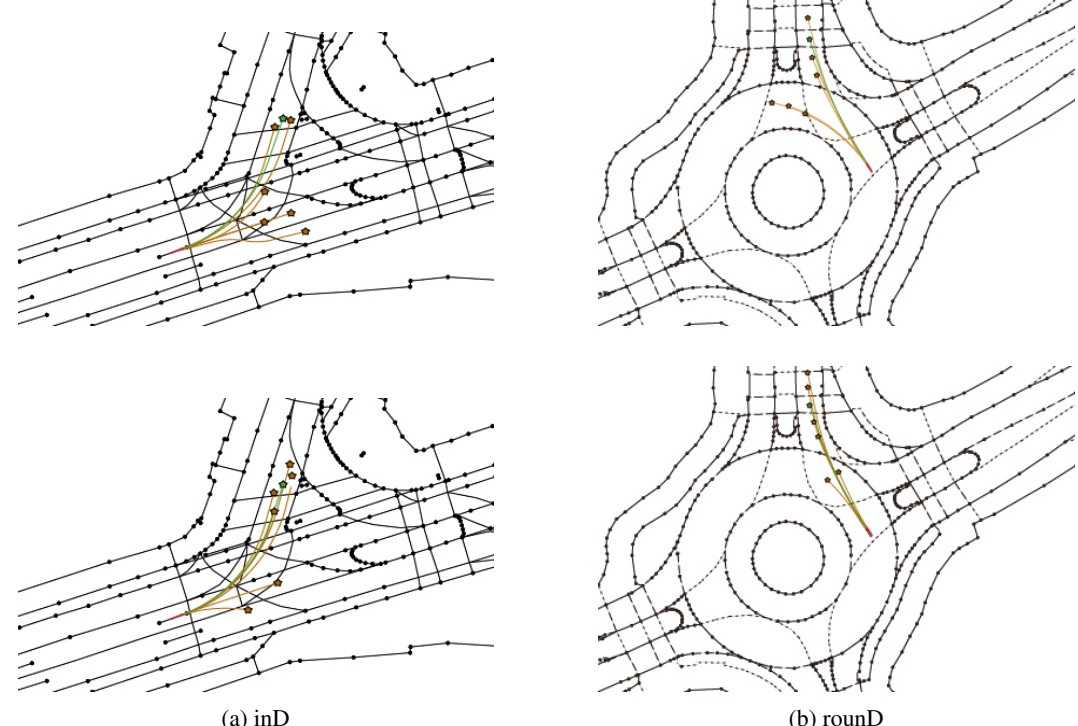

(a) inD                                    (b) rounD

Figure 3: Qualitative analysis. The first row shows the visualization results of HPNet, while the second row shows the visualization results of our method.

optimizer was employed with a batch size of 64, and the model was trained for 50 epochs. The learning rate was linearly decayed from $10^{-3}$ to 0.

## 4.2 EXPERIMENTAL RESULTS

**Comparison with baselines.** We compare the performance of LSTR with map-rich methods on the inD and RounD datasets, including DenseTNT, HOME , GANet , Wayformer , ProphNet , QCNet , and HPNet as shown in Table 1. Our method outperforms all other approaches across all metrics on both datasets, demonstrating LSTR's effectiveness in weakly constrained environments like urban intersections and roundabouts. Figures 3(a)(b) display the qualitative results on the inD and RounD datasets. Notably, greedy models based on strong rules perform poorly on multimodal metrics compared to our global approach, likely because they do not directly optimize for long-range global intentions but rely on temporal dependencies under road constraints. This further shows that our model effectively captures vehicles' route choice patterns.

**Comparison with State-of-the-art.** As noted in related works, the MTR series are strong map-prior algorithms, showing impressive performance on the Waymo Open Motion Dataset leaderboard. To further evaluate our method in complex intersections, we compared it with MTRv3 on several WOMD intersection scenes. For fairness, we disabled intersection map features in WOMD from the Backbone of both models. The results are shown in the Table 2. Our method outperforms MTRv3 by 5.1% in $minADE_6$, as the absence of map features hinders the offline search for multimodal intention queries, especially for difficult scenarios like turns and U-turns, limiting the refinement of local movement. Our method also slightly outperforms MTRv3 on other metrics.

## 4.3 ABLATION STUDIES

**Effectiveness of Short-range Trajectory Prediction.** The introduction of the Local Behavior Decision Module (LBDM) improved the model's performance across multiple evaluation metrics. As shown in the Table 3, adding LBDM reduced $b\text{-}minFDE_6$ from 2.429 to 2.264, and $minADE_6$ im-

| | Method | b-minFDE$_6$ | minADE$_6$ | minFDE$_6$ | MR$_6$ | minADE$_1$ | minFDE$_1$ | MR$_1$ |
|---|---|---|---|---|---|---|---|---|
| **inD** | DenseTNT | 2.146 | 0.655 | 1.354 | 0.116 | 1.481 | 3.585 | 0.504 |
| | HOME | 2.075 | 0.627 | 1.309 | 0.109 | 1.426 | 3.518 | 0.478 |
| | GANet | 2.015 | 0.605 | 1.276 | 0.103 | 1.370 | 3.453 | 0.455 |
| | WayFormer | 1.964 | 0.580 | 1.241 | 0.097 | 1.323 | 3.400 | 0.440 |
| | QCNet | 1.904 | 0.569 | 1.213 | 0.093 | 1.283 | 3.352 | 0.430 |
| | HPNet | 1.901 | 0.567 | 1.210 | 0.092 | 1.280 | 3.347 | 0.428 |
| | **LSTR (Ours)** | **1.815** | **0.538** | **1.173** | **0.087** | **1.256** | **3.322** | **0.421** |
| **rounD** | DenseTNT | 2.718 | 0.938 | 1.350 | 0.135 | 2.081 | 3.987 | 0.528 |
| | HOME | 2.625 | 0.915 | 1.290 | 0.128 | 2.025 | 3.925 | 0.500 |
| | GANet | 2.541 | 0.893 | 1.233 | 0.120 | 1.964 | 3.859 | 0.475 |
| | WayFormer | 2.475 | 0.875 | 1.180 | 0.115 | 1.912 | 3.803 | 0.457 |
| | HPNet | 2.393 | 0.850 | 1.151 | 0.107 | 1.868 | 3.753 | 0.441 |
| | QCNet | 2.391 | 0.849 | 1.150 | 0.108 | 1.870 | 3.755 | 0.440 |
| | **LSTR (Ours)** | **2.258** | **0.814** | **1.106** | **0.105** | **1.833** | **3.717** | **0.434** |

Table 1: Comparison results on inD and rounD sorted by b-minFDE$_6$. The best entry for a metric is marked bold, and the second best is underlined.

| Method | minADE$_6$ | minFDE$_6$ | MR$_6$ |
|---|---|---|---|
| MTRv3[†] | 0.764 | 1.559 | 0.167 |
| **LSTR(Ours)** | **0.725** | **1.534** | **0.165** |

Table 2: Comparison resultes between our method and MTRv3 in several intersection scenarios on the WOMD dataset. [†]:Since MTRv3 has not published a paper or released its source code, we implemented it based on the technical report, using MTR as the foundation. In our version, we omitted the Lidar input but retained the offline search component.

proved from 0.731 to 0.679. A decrease in minFDE$_6$ and MR$_6$ was also observed, indicating that LBDM effectively captures short-range motion patterns and enhances local decision-making for better trajectory prediction.

**Effectiveness of Global Intention Guidance.** The inclusion of the Global Intention Inference Module (GIIM) significantly enhanced the model's performance. When combining the GIIM module with LBDM, b-minFDE$_6$ further decreased from 2.264 to 2.181, and minADE$_6$ dropped from 0.679 to 0.648. Even when GIIM is incorporated into the decoder without LBDM, minFDE$_6$ improved from 1.499 to 1.226, demonstrating that this module is essential for capturing the global intentions of vehicles. When GIIM is used with the multi-modal decoder, all evaluation metrics significantly improved, proving the effectiveness of global intention inference in enhancing trajectory prediction accuracy.

**Effectiveness of the Multi-modal Decode.** The multi-modal decoder achieved the best overall performance. Even when used alone, it reduced b-minFDE$_6$ from 2.429 to 2.387. Combined with LBDM and GIIM, b-minFDE$_6$ dropped to 1.815, minADE$_6$ to 0.538, and minFDE$_6$ to 1.173, marking LSTR's best results on the inD dataset. This highlights the decoder's effectiveness in capturing global intentions and local decision patterns.

## 5 CONCLUSION

In this paper, we proposed an advanced trajectory prediction model for vehicles at urban intersections, integrating local behavior decision-making, global intention inference, and multi-modal decoding. Our model effectively captures both short-term and long-term dynamics, improving the accuracy and robustness of predictions. The experimental results demonstrate that each component contributes significantly to reducing error metrics, with the full model achieving state-of-the-art performance. This work highlights the importance of modeling both local interactions and global

| Module Setup | | | Final evaluation metrics | | | |
|---|---|---|---|---|---|---|
| LBDM | GIIM | Decoder | b-minFDE$_6$ | minADE$_6$ | minFDE$_6$ | MR$_6$ |
| | | | 2.429 | 0.731 | 1.546 | 0.130 |
| ✓ | | | 2.264 | 0.679 | 1.433 | 0.117 |
| ✓ | ✓ | | 2.181 | 0.648 | 1.380 | 0.111 |
| | | ✓ | 2.387 | 0.724 | 1.499 | 0.128 |
| ✓ | | ✓ | 2.034 | 0.612 | 1.329 | 0.099 |
| | ✓ | ✓ | 1.899 | 0.554 | 1.226 | 0.090 |
| ✓ | ✓ | ✓ | **1.815** | **0.538** | **1.173** | **0.087** |

Table 3: The final performance of all metrics with $k = 6$ for various LSTR module combinations on the inD dataset is provided. The standalone GIIM module is not evaluated.

intentions, providing a solid foundation for future research in autonomous driving systems, particularly in complex and dynamic urban environments.

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

# A  APPENDIX

## A.1  METRIC DETAILS

**Average Displacement Error (ADE).** calculates the average $L^2$-norm of the error between predicted positions $\hat{x}_k$ and ground truth positions $x_k$ over the entire prediction horizon.

$$\text{ADE} = \frac{1}{N} \sum_{k=1}^{N} \|\hat{x}_k - x_k\|_2 \tag{6}$$

Where, $N$ is the number of predicted points, $\hat{x}_k$ is the predicted position at time step $k$, $x_k$ is the ground truth position, $\|\cdot\|_2$ denotes the $L^2$-norm, i.e., the Euclidean distance between predicted and true positions. And Minimum ADE (minADE$_K$) refers to the minimum ADE over $K$ predictions, where the best (smallest) error is selected.

**Final Displacement Error (FDE).** measures the $L^2$-norm (Euclidean distance) between the final predicted position $\hat{x}_N$ and the ground truth $x_N$. It indicates the model's accuracy in predicting distant future events.

$$\text{FDE} = \|\hat{x}_N - x_N\|_2 \tag{7}$$

Where, $\hat{x}_N$ is the predicted final position at time step $N$, $x_N$ is the ground truth final position, $\|\cdot\|_2$ is the $L^2$-norm, i.e., the Euclidean distance between the predicted and true final positions. Minimum

FDE (minFDE$_K$) refers to the minimum FDE over $K$ predictions, selecting the smallest error in the final predicted position.

**Miss Rate (MR).** The miss rate (MR) refers to the proportion of cases where the predicted final position is more than 2 meters from the ground truth.

**Brier Final Displacement Error (b-FDE).** is a variation of Final Displacement Error (FDE) that incorporates the probability of each predicted mode, useful for multimodal prediction tasks.

$$\text{Brier-FDE} = \left(1 - \pi^j\right)^2 \|\hat{x}_N^j - x_N\|_2 \tag{8}$$

where $\pi^j$ is the predicted probability of the $j$-th mode, $\hat{x}_N^j$ is the predicted final position, and $x_N$ is the ground truth. $\|\cdot\|_2$ is the $L^2$-norm (Euclidean distance). The weighting factor $\left(1 - \pi^j\right)^2$ penalizes errors based on predicted probability, giving more weight to higher probability predictions. And b-minFDE minimizes the final displacement error by selecting the trajectory whose final position is closest to the ground truth.

## A.2 DATASET PROCESSING

Downsampling trajectory data speeds up prototyping and experimentation, but excessively low sampling rates may distort trajectories. To balance this, we downsampled the inD and RounD datasets to 10 Hz (0.1-second intervals) while preserving the original information. Analysis revealed components exceeding the new Nyquist frequency (5 Hz), so a 7th-order Chebyshev Type I filter was applied before downsampling to remove high-frequency components.

For the inD dataset, 8,233 vehicle trajectories from four intersections were split into training, validation, and test sets (0.8:0.1:0.1). We predicted future trajectories using a 3-second observation window and a 5-second prediction horizon. For the RounD dataset, all 13,509 vehicle trajectories from three roundabouts were used for training, with the same data splits and observation/prediction windows.

To validate generalizability, we used 60 intersection scenarios from the Waymo Open Motion Dataset, comprising about 580,000 data points from 6,640 vehicles. Time steps were 0.1 seconds apart, and the data was split (0.8:0.1:0.1), using the same observation and prediction windows as above.

## A.3 FINE-GRAINED PERFORMANCE EVALUATION ON SPECIFIC DRIVING MANEUVERS

In Table 1 of the main paper, we present a comparison of the performance of various trajectory prediction models. This Table 4 presents a comparison of different models' predictive performance under fine-grained maneuvers using the Waymo Open Motion Dataset. Distinct driving behaviors (such as left turns, right turns, and U-turns) exhibit markedly different motion patterns. By evaluating model performance in these specific behaviors, we gain deeper insights into the model's generalization ability, helping to identify potential weaknesses and ensure robustness in complex scenarios such as intersections.

| Traj. Type | Stationary | Straight | Right u-turn | Right-turn | Left u-turn | Left-turn |
|---|---|---|---|---|---|---|
| MTRv3 | 1.32 | 1.52 | 5.89 | 2.68 | 3.10 | 2.45 |
| LSTR(Ours) | 1.35 | 1.54 | 5.51 | 2.53 | 3.08 | 2.24 |

Table 4: Comparison of brier-minFDE$_6$ results for trajectory prediction across different maneuver types between LSTR (Ours) and MTRv3.

The table presents a comparison of LSTR (Ours) and MTRv3 in terms of brier-minFDE$_6$ across various trajectory types. For key turning scenarios, such as right U-turns and right turns, LSTR exhibited superior performance, reducing the error by 6.5% and 5.6%, respectively. Although MTRv3 slightly outperforms LSTR in stationary and straight scenarios, the margin remains within 2%. The two methods perform similarly in left U-turn and left turn scenarios, with LSTR maintaining a slight advantage. This indicates that LSTR effectively enhances prediction accuracy in complex turning scenarios while retaining overall competitiveness.

## A.4 Comparison of Coordinate Systems for Intersection Trajectory Prediction

In intersection trajectory prediction, the selection of a coordinate system plays a critical role in determining the accuracy and efficiency of the prediction model. Early works intuitively considered using a global coordinate system to capture vehicle motion. However, a potential issue with the global coordinate system is that it may promote prediction bias, where the model learns patterns specific to certain scenes instead of generalizable behavior models—an inherent challenge in zero-shot learning. The table 5 presents the test results of our method under different coordinate systems on the inD dataset and Waymo Open Motion Dataset:

| Coordinate System | b-minFDE$_6$ | minADE$_6$ | minFDE$_6$ | MR$_6$ | minADE$_1$ | minFDE$_1$ | MR$_1$ |
|---|---|---|---|---|---|---|---|
| Scene-centric | 1.90 | 0.56 | 1.21 | 0.10 | 1.30 | 3.41 | 0.44 |
| Agent-centric | 1.82 | 0.54 | 1.17 | 0.09 | 1.26 | 3.32 | 0.42 |

Table 5: The test results of our method under inD.

| Coordinate System | minADE$_6$ | minFDE$_6$ | MR$_6$ |
|---|---|---|---|
| Scene-centric | 0.77 | 1.56 | 0.18 |
| Agent-centric | 0.73 | 1.53 | 0.17 |

Table 6: The test results of our method under WOMD

