# OpenReview forum: "LSTR: Long-Short Range Aggregation for Trajectory Prediction at Intersection Scenarios"
_ICLR.cc/2025/Conference — Submitted to ICLR 2025_

### Official Review · Reviewer_Jr6q · 2024-11-01

**Soundness:** 3
**Presentation:** 3
**Contribution:** 3
**Rating:** 6
**Confidence:** 4

**Summary:**

This paper introduces LSTR, a trajectory prediction model designed for complex urban intersections. . LSTR combines global intention inference with local interaction modeling, capturing diverse motion patterns.

**Strengths:**

* Overall the paper is well-written and easy to follow.
* The proposed method study an interesting while relatively less explored problem for how to model the global/local intents under no or sparsely annotated lane information.
* The author benchmarked the performance on multiple datasets including the inD,rounD, and WOMD.

**Weaknesses:**

* The main benchmark on WOMD only compared with the MTRv3. More comparisons with other SOTA methods in WOMD are needed to justify the technical improvement especially given MTRv3's result is based on author's reimplementation. More open-sourced methods can be used for more fair comparison.
* In the motivation, the author claimed that existing methods would fail in intersections without detailed annotations. However, in the experiment section, it seems all the selected intersections/dataset have detailed map annotations as shown in Figure 3 and the proposed method would also consume lane graph as shown in Figure 2. Key questions like whether the proposed method would outperform other SOTA methods under scenarios with no detailed map annotations are still not fully justified.

**Questions:**

* There are some minor typos in the paper. Please correct it e.g., "flows Zhao et al. (2023).Microscopic traffic flow" with extra '.' and also should add extra space after it.
* What is the criteria for selecting subset of WOMD?

---

### Official Review · Reviewer_kt2g · 2024-11-02

**Soundness:** 2
**Presentation:** 2
**Contribution:** 1
**Rating:** 3
**Confidence:** 5

**Summary:**

The paper introduces LSTR, a model that enhances vehicle trajectory prediction in complex urban intersections. It combines local interaction modelling with long-range intention prediction to handle chaotic intersection traffic. It adapts predictions to individual vehicle dynamics and overarching traffic flow using modules for local behaviour decisions and global intention inference.

**Strengths:**

By integrating local behavior decision-making with global intention inference, LSTR effectively captures both short-range vehicle dynamics and long-range traffic patterns, improving prediction accuracy in chaotic intersection environments. Extensive experiments on datasets like inD, rounD, and WOMD confirm LSTR’s superior performance in accurately predicting diverse motion patterns in real-world intersection scenarios.

**Weaknesses:**

Please refer to the Questions section.

**Questions:**

1. Intro: The challenges and motivations in the introductory section are inadequately supported. While the authors claim that most trajectory prediction methods rely on high-definition maps, they fail to discuss numerous recent map-free approaches.

2. The core challenges highlighted are somewhat ambiguous, lacking references to substantiate the claims. The authors seem to assert these points subjectively without justifying why prior work has not addressed these issues or why these challenges are critical to trajectory prediction. Moreover, terms like local interaction/global interaction/ and route choice pattern lack standardized definitions in this field, leading to confusion and reducing readability.

3. Terms such as short-range motion patterns/long-term dynamics/anchors global intentions lack clear definitions. While anchors are used in computer vision (e.g., DETR), it’s unusual in autonomous trajectory prediction, making it hard to interpret here. It appears the authors themselves may not fully understand these terms. Using complex terminology without clear explanations can hinder the paper’s readability.

4. The paper’s contributions are not sufficiently novel, as evidenced by the limited number of 2024 references, with most citations being relatively outdated. Additionally, the related work section should include map-free methods to present a comprehensive overview of the field.

5. There are several formatting errors, such as misplaced quotation marks (e.g., lines 41, 52, 80, and 89) and missing spaces after numbers (lines 85, 89, 92). These errors detract from the manuscript’s professionalism.

6. In the Approach section, the authors do not specify the map information being used. Moreover, the authors initially criticize other methods for over-relying on map-based information, yet they use map information in their approach, leading to inconsistency in their argument.

7. The proposed methodology seems like a case of "fitting the solution to the problem" giving the impression of a model assembled without genuine innovation. The approach appears to be a combination of ideas from HiVT (CVPR ‘22) and QCNet (CVPR ’24), with no substantial distinction from previous works.

8. The authors should clarify whether the rounD dataset includes detailed map information. As far as I recall, no such detailed maps are provided, given that data collection was done via drones.

9. The experimental section states, “We use the official Argoverse competition metrics” which is confusing since the experiments were conducted on three different datasets (inD, rounD, WOMD). Why use metrics from an unrelated dataset? Are there no suitable metrics for the chosen datasets?

10. Certain acronyms and technical terms lack definition, such as "VRUs" (Vulnerable Road Users, line 304) and formatting artefacts like the bolding of “b-minFDEk” (line 320). These should be properly introduced and formatted.

11. The paper appears to be heavily reliant on LLMs for drafting. While such tools can enhance readability, phrases often seem vague or verbose, indicating a lack of manual refinement. This results in an awkward, overly complex writing style.

12. The contributions claim that the model is capable of making interpretable predictions, but there is no substantial evidence or detailed explanation to support this claim, which is a significant shortcoming.

13. Given my familiarity with HiVT, QCNet, and HOME, transferring these architectures to datasets like inD and RounD seems challenging. I am sceptical of the validity of the results and suggest conducting experiments on benchmark datasets like Argoverse or NuScenes to further substantiate the model’s performance.

14. The baseline models used in the comparison are relatively old. More recent baselines should be included to provide a robust evaluation of the model’s performance.

15. The paper lacks ablation studies and a Discussion section on the model's limitations and potential future work, both of which are essential to understanding the model’s robustness and areas for improvement.

16. The qualitative analysis figures (e.g., Fig. 3) are unclear, making it difficult to distinguish ground truth from model predictions. Additionally, visualizing model performance using WOMD’s official API could provide clearer insights into performance in complex scenarios.

Overall, the manuscript does not meet publication standards. Significant revisions are required to address the issues highlighted above, particularly regarding conceptual clarity, experimental thoroughness, and the novelty of contributions.

---

### Official Review · Reviewer_kAdX · 2024-11-04

**Soundness:** 2
**Presentation:** 2
**Contribution:** 1
**Rating:** 3
**Confidence:** 4

**Summary:**

The paper highlights the two core challenges of intersection trajectory prediction: local interaction and global intention. Correspondingly, the paper proposes a framework, LSTR, which consists of the Local Behavior Decision Module (LBDM), the Global Intention Inference Module (GIIM), and a final decoder to overcome these challenges. The proposed LSTR surpasses state-of-the-art methods in intersection scenarios; ablations demonstrate the effectiveness of the proposed modules.

**Strengths:**

1. The paper highlights the two core challenges of trajectory prediction in intersection scenarios: local interaction and global intention.

2. The proposed LSTR surpasses state-of-the-art methods in intersection scenarios. Ablations demonstrate that the major modules in LSTR: the Local Behavior Decision Module (LBDM), the Global Intention Inference Module (GIIM) and a final decoder are effective.

3. The paper is easy to follow, with a clear problem definition, equations, and figures for enhancing the reader's understanding.

**Weaknesses:**

1. The motivation and contributions of this paper are unclear to me. The proposed LSTR framework integrates several existing techniques, such as encoders in MTR [1] and HiVT [2], and the offset predictor in QCNet [3], except for pattern re-featuring and map filtering in GIIM, which are described only briefly. Note that papers [1] and [2] aim to address challenges of local interaction and global intention in general scenarios. My major concerns are: (1) Why do intersection scenarios need a custom framework? (2) How do the differences between the proposed LSTR and existing methods contribute to addressing the aforementioned challenges, especially in intersection scenarios?

2. The paper lacks implementation details of LSTR, such as hyperparameters like the hidden size, the number of layers in each module, and the radius used to collect information about agents and maps in scenarios.

3. There are several errors in the paper: the caption for the legend in Fig. 1 is missing. There are typos, such as those on lines 71 and 282. There is also a question regarding the related works section: Does MmTransformer [4] belong to the MTR [1] series?

[1] Shi et al. Motion Transformer with Global Intention Localization and Local Movement Refinement. NeurIPS 2022. \
[2] Zhou et al. HiVT: Hierarchical Vector Transformer for Multi-Agent Motion Prediction. CVPR 2022. \
[3] Zhou et al. Query-Centric Trajectory Prediction. CVPR 2023. \
[4] Liu et al. Multimodal Motion Prediction with Stacked Transformers. CVPR 2021.

**Questions:**

1. See weaknesses.

2. What are the parameter size and inference latency of LSTR compared to the baselines?

---

### Official Review · Reviewer_jveU · 2024-11-04

**Soundness:** 3
**Presentation:** 3
**Contribution:** 3
**Rating:** 6
**Confidence:** 3

**Summary:**

The Long-short Range Aggregation for Trajectory Prediction in Intersection (LSTR) model adeptly handles the macroscopic organization and microscopic disorder of urban intersection traffic flows. It enhances short-range motion pattern prediction through the Coherent Occupancy Prediction (COP) head, facilitating the parallel forecasting of future trajectories and effectively capturing local dynamics. The model also includes a Global Intention Inference Module (GIIM) for destination prediction and the integration of global intentions with local decisions. Emphasizing its superiority over map-prior algorithms through integrating local interaction modeling with global intention prediction, LSTR demonstrates significant performance improvements on datasets, with a notable 4.5 improvement in b-minFDE6 on inD and a 4.2 improvement in minADE6 on rounD over the next best method. These results underscore LSTR's enhanced accuracy in trajectory prediction for complex intersection scenarios.

**Strengths:**

The paper introduces a novel approach to trajectory prediction at complex intersections and roundabouts, which is inherently an area ripe for innovation. The Long-short Range Aggregation for Trajectory Prediction in Intersections (LSTR) model distinguishes itself by focusing on the critical elements of trajectory entry and exit points within the Macroscopic Traffic Flow Generation module. This design choice, coupled with the sequential refinement of local path planning followed by global exit point-based trajectory tuning, presents a unique methodology that deviates from mainstream approaches. The model's architecture and the strategic integration of local and global intentions exhibit a creative synthesis of existing concepts, thereby fulfilling the broad definition of originality in research.

**Weaknesses:**

1）The approach of utilizing COP in conjunction with self-attention within the LBDM for trajectory forecasting may not be considered a significant innovation. The method's reliance on historical data to directly predict future states does not present a groundbreaking advancement in the field of trajectory prediction. Subsequent iterations of the model could incorporate more complex predictive algorithms or integrate additional data sources to enhance its innovativeness and predictive accuracy.

2）The paper would benefit from a detailed disclosure of the model's parameter count, which is crucial for assessing the computational efficiency and practicality of the LSTR model. A comparative analysis of parameter volume with other models would provide a clearer understanding of how the model scales and performs relative to existing solutions in the domain.

3）The placement of Table 3 within the Conclusion section appears to be misplaced, which may disrupt the logical flow of the paper. A more appropriate section for this table would be one dedicated to results or discussion to maintain the coherence of the manuscript. Furthermore, the appendices should maintain a consistent referencing style for tables and figures to uphold the scholarly standards of the paper. The use of specific table references, such as "Table 1," should be adopted uniformly to improve the academic rigor and presentation quality of the work.

**Questions:**

1、	LBDM is a critical component of the LSTR model,  it seems that it may not significantly deviate from existing prediction methods. Could the authors articulate the distinct innovative elements of the LBDM that set it apart from existing trajectory prediction methods?

2、	The paper mentions an experimental condition where intersection map features were disabled in the Backbone of both the LSTR model and comparative models for fairness. Why were map features disabled in the experiments, and how does this affect the model's reliance on environmental information?

3、	What adaptations could be made to the LSTR model to improve its performance in scenarios with a high presence of Vulnerable Road Users (VRUs)? How might the model be adapted to improve its predictive accuracy in such environments, and what additional data or model components might be necessary to achieve this?

---

### Meta-Review · Area_Chair_yEgY · 2024-12-20

**Metareview:**

This work proposes a novel trajectory predictor that involves reasoning about long-range intentions. While the problem is interesting and relevant, the formulation work is formulated in a way that was considered by some reviewers as unclearly motivated and ambiguous at times in addition to several further shortcomings. Ultimately, the rejection decision is based on the fact that none of the (many) shortcomings raised by the reviewers have been addressed.

**Additional Comments On Reviewer Discussion:**

The authors did not submit a rebuttal.

---

### Decision · Program_Chairs · 2025-01-22

Reject